# Sport and dance interventions for healthy young people (15–24 years) to promote subjective well-being: a systematic review

Louise Mansfield,[1] Tess Kay,[2] Catherine Meads,[3] Lily Grigsby-Duffy,[2] Jack Lane,[4] Alistair John,[2] Norma Daykin,[5] Paul Dolan,[6] Stefano Testoni,[6] Guy Julier,[4] Annette Payne,[2] Alan Tomlinson,[4] Christina Victor[7]

[1]Department of Life Sciences, Division of Sport Health and Exercise Sciences, Brunel University London, Uxbridge, UK
[2]Life Sciences, Brunel University London, Uxbridge, UK
[3]Health, Social Care and Education, Anglia Ruskin University, Cambridge, UK
[4]Arts and Humanities, University of Brighton, Brighton, UK
[5]Health and Wellbeing, University of Winchester, Winchester, UK
[6]Social Policy, The London School of Economics and Political Science, London, UK
[7]Clinical Sciences, Brunel University London, Uxbridge, UK

**Correspondence to**
Professor Louise Mansfield;
louise.mansfield@brunel.ac.uk

## ABSTRACT

**Objective** To review and assess effectiveness of sport and dance participation on subjective well-being outcomes among healthy young people aged 15–24 years.

**Design** Systematic review.

**Methods** We searched for studies published in any language between January 2006 and September 2016 on PsychINFO, Ovid MEDLINE, Eric, Web of Science (Arts and Humanities Citation Index, Social Science and Science Citation Index), Scopus, PILOTS, CINAHL, SPORTDiscus and International Index to Performing Arts. Additionally, we searched for unpublished (grey) literature via an online call for evidence, expert contribution, searches of key organisation websites and the British Library EThOS database, and a keyword Google search. Published studies of sport or dance interventions for healthy young people aged 15–24 years where subjective well-being was measured were included. Studies were excluded if participants were paid professionals or elite athletes, or if the intervention was clinical sport/dance therapy. Two researchers extracted data and assessed strength and quality of evidence using criteria in the What Works Centre for Wellbeing methods guide and GRADE, and using standardised reporting forms. Due to clinical heterogeneity between studies, meta-analysis was not appropriate. Grey literature in the form of final evaluation reports on empirical data relating to sport or dance interventions were included.

**Results** Eleven out of 6587 articles were included (7 randomised controlled trials and 1 cohort study, and 3 unpublished grey evaluation reports). Published literature suggests meditative physical activity (yoga and Baduanjin Qigong) and group-based or peer-supported sport and dance has some potential to improve subjective well-being. Grey literature suggests sport and dance improve subjective well-being but identify negative feelings of competency and capability. The amount and quality of published evidence on sport and dance interventions to enhance subjective well-being is low.

**Conclusions** Meditative activities, group and peer-supported sport and dance may promote subjective well-being enhancement in youth. Evidence is limited. Better designed studies are needed.

**Trial registration number** CRD42016048745; Results.

## Strengths and limitations of this study

► Prepublication of our protocol on the International Prospective Register of Systematic Reviews ensures methodological transparency and mitigates against potential post hoc decision making.
► A comprehensive research, policy and practice-relevant search strategy was used including searches of published and unpublished data, and study selection was carried out by two reviewers independently.
► Data extraction and quality assessments were conducted using standardised forms, independently by two reviewers.
► The focus on a specific target age group may have excluded evidence from studies that have aggregated data across younger and older age groups in their analysis.
► Meta-analysis was not possible due to the heterogeneity of study interventions and outcomes.

## INTRODUCTION

Governments and international organisations acknowledge subjective well-being (SWB) as a policy goal.[1–3] The international focus on measuring SWB is gaining recognition in some aspects of UK sport,[4 5] dance[6] and physical activity policy.[7] SWB describes well-being in terms of the good and bad feelings arising from what people do and how they think.[8] Good feelings include happiness, joy, contentment and excitement while sadness, worry, stress and anxiety are examples of more negative feelings. People's experiences also involve a sense of purpose (eg, worthwhileness, meaningfulness) and pointlessness (eg, futility, boredom). Since 2011, SWB measured as satisfaction with life, worthwhileness, happiness and anxiety has been included in UK population surveys conducted by the Office of National Statistics.[9] Links between sports and cultural activities

and SWB have been reported and sport engagement is included in national-level data collection and analysis.[10] Significant associations have been found between engagement in sport, the arts and enhanced SWB as measured by life satisfaction.[11] Yet, it is acknowledged that SWB is a complex concept, with no single agreed definition or measure.[12] The term SWB is used synonymously with a wide range of concepts including self-esteem, self-efficacy, self-determination, resilience, quality of life, mood enhancement, positive mental health, life satisfaction, worthwhileness and happiness.[13] Measures of SWB use various scales that demonstrate well-being as multidimensional, for example, The Warwick and Edinburgh Mental Wellbeing Scale,[14] Rosenberg's Self-Esteem Scale and[15] The Profile of Mood States.[16]

The What Works Centre for Well-being initiative,[17] funded by the Economic and Social Research Council has commissioned evidence reviews in key areas including Culture, Sport and Well-being. Following consultation with stakeholders,[18] four topics were identified for systematic review between 2015 and 2018 (music and singing, sport and dance, visual arts, and outdoor nature-based physical activity). This paper reports the findings of the second systematic review topic; sport and dance interventions for healthy young people (15–24 years) to promote subjective well-being.

The established definition of sport, used throughout the sector, remains that cited from the European Charter (1992) and refers to forms of physical activity either casually or formally organised in which people take part for fitness, health and well-being, social relationships or competition.[19] Sport includes a wide range of individual and group activities including jogging, running, cycling, martial arts, yoga, team games and athletics. Dance is commonly defined differently from sport as a performing art form which refers to the rhythmic movements and sequences of steps usually set to music.[20] Sport and dance programmes in the UK operate in different delivery, programming and funding environments, yet both sport and dance organisations identify young people as a key target group for engagement in physical activity to enhance well-being. The evidence, however, is theoretically and methodologically diverse and less attention has been given to children and adolescents. Existing evidence reviews on sport have tended to focus on physical rather than mental health or well-being outcomes[21–23] or they have examined the effect of exercise in populations with specific mental health conditions such as depression[24] and anxiety.[25 26] Dance-related reviews of evidence have examined the effectiveness of dance therapy on psychological and physical health and well-being outcomes in patients with cancer,[27] for schizophrenia[28] and on depression.[29] A review of reviews on physical activity and mental health in children and adolescents identified an association between physical activity and positive well-being outcomes connected to reduced depression and anxiety, and enhanced self-esteem and cognitive function.[30] No systematic review to date has focused on sport

and dance interventions in healthy young people (15–24 years) to promote subjective well-being. Interventions that positively influence the well-being of young people have the potential to promote good physical and mental health.[31–33] This review provides evidence that may improve understanding of the effects of sport and dance on a range of SWB measures and contribute to informing policy development, programme delivery and measurement and evaluation of sport and dance interventions to enhance well-being.

## METHODS

The protocol for this systematic review was registered with the International Prospective Register of Systematic Reviews on 3 October 2016 (registration number CRD42016048745). The review follows the Preferred Reporting Items for Systematic Reviews and Meta-Analysis guidelines.[34]

### Patient and public involvement

Participant observation by one investigator (LM) of public groups taking part in community arts and sports activities contributed to the development of the review question for this study. Patients were not involved in the conduct of the systematic review. The findings of this study will be written in accessible English and disseminated through the What Works Centre for Wellbeing website accessible by the public.

### Inclusion criteria

Inclusion criteria were any comparative studies investigating any form of sport or dance compared with no sport or dance, usual routine or comparing pretest and posttest scores in healthy young people aged 15–24 years and measuring any form of subjective well-being (table 1). We included studies worldwide from countries economically similar to the UK (using OECD–DAC list of country development; http://www.oecd.org/dac/stats/daclist.htm) or with study populations similar in terms of socioeconomic status. Studies could be fully published with search dates of 2006–2016 to reflect current and long-term work on sport, dance and well-being,or grey literature (with search dates of 2013–2016). Shorter timescales for grey literature search ensured a focus on finding recent relevant studies that had not yet been published. Grey literature was included if it was a final evaluation or report on empirical data, had the evaluation of sport-related or dance interventions as the central objective and included details of authors (individuals, groups or organisations).

### Exclusion criteria

Published studies were excluded if participants were paid professionals or elite athletes, or if the intervention was sport or dance therapy delivered in a clinical setting for rehabilitation purposes. We did not include studies of walking as there is existing review level evidence on the health and well-being benefits of this activity.[35 36] Grey

**Table 1** Eligibility criteria for selecting studies

| PICOS criteria | Inclusion | Exclusion |
|---|---|---|
| Participants | ▶ Participants were to be 15–24 years of age.<br>▶ Studies from countries economically similar to the UK (ie, other high-income countries with similar economic systems) or with study populations that have similar socioeconomic status to the UK. | ▶ Participants with a health condition diagnosed by a health professional.<br>▶ Participants who were paid professionals or elite athletes.<br>▶ Participants in clinically based sport and dance interventions. |
| Intervention | ▶ Participatory sport and dance interventions including watching and performing.<br>▶ Including sport-related and dance therapy offered to enhance well-being in healthy young people. | ▶ Clinical sport-based or dance therapy.<br>▶ Sport and dance for clinical procedures such as surgery, medical tests and diagnostics.<br>▶ Walking. |
| Comparison | ▶ No sport or dance, usual routine, ie, inactive comparator or historical/time-based comparator, ie, pre-post study design. | |
| Outcomes | ▶ Subjective well-being using any recognised method or measure. | |
| Study design | ▶ Empirical research: either quantitative, qualitative or mixed methods, outcomes or process evaluations.<br>▶ Grey literature: if it was a final evaluation or report on empirical data, had the evaluation of sport-related or dance interventions as the central objective and included details of authors (individuals, groups or organisations).<br>▶ Published studies published between 2006 and 2016.<br>▶ Grey literature and practice surveys published between 2013 and 2016. | ▶ Discussion articles, commentaries or opinion pieces not presenting empirical or theoretical research.<br>▶ Grey literature if it did not have details of authorship. |

PCOS, Population Intervention Comparator Outcome Study Design.

literature was excluded if it did not meet the criteria for inclusion on date, format of reporting, type of data and details of authorship. Eligibility criteria are summarised in table 1.

### Data sources and search strategy

We searched for empirical studies published between January 2006 and September 2016 on the following databases: PsychINFO, Ovid MEDLINE, Eric, Web of Science (Arts and Humanities Citation Index, Social Science and Science Citation Index), Scopus, PILOTS, CINAHL, SPORTDiscus and International Index to Performing Arts. There were no language restrictions.

Electronic databases were searched using a combination of Medical Subject Headings (MeSH) and free text terms. An example of the Ovid MEDLINE search strategy used can be found in online supplementary appendix 1. All database searches were based on this strategy but were appropriately revised to suit each database.

Additionally, reference lists of all relevant reviews[37–42] from the last 5 years were hand-searched to identify additional relevant empirical evidence. We also carried out a search for up-to-date UK unpublished (grey) literature completed between 2013 and 2016 via: (i) an online call for evidence on the What Works Centre for Wellbeing website between October and November 2016; (ii) contacting known experts in the field for recommendations of sport or dance sector reviews or repositories to search; (iii) a review of key sector websites; (iv) a search of the British Library EThOS website for unpublished PhD dissertations and (v) reviewing the titles of the first 100 results in a Google search with the use of key terms ('sport' AND 'physical activity' AND 'dance' AND 'wellbeing' AND 'young people'). 'Physical activity' was included as a search term because it is used by the sector when reporting on sport and dance activities.

### Study selection

Two reviewers independently screened the titles and abstracts of all studies identified by the search strategy for their eligibility. Where it was not clear from the title and abstract whether a study was relevant, the selection criteria were independently applied to the full article to confirm its eligibility. Where two independent reviewers did not agree in their primary judgements they discussed the conflict and attempted to reach a consensus. If they could not agree then a third member of the review team considered the full paper and a majority decision was made. Online supplementary appendix 2 lists excluded studies and reasons for exclusions.

## Data extraction

Two review authors independently extracted data using a standardised form (online supplementary appendix 3). Discrepancies were resolved by discussion and consensus. Where agreement could not be reached, a third review author considered the paper and a majority decision was reached. The following data were extracted: (1) evaluation design and objectives (the interventions studied and control conditions used, including detail where available on the intervention content, dose and adherence, ethics); (2) sample (size, representativeness, reporting on dropout, attrition and details of participants including demographics and protected characteristics where reported); (3) the outcome measures (the scales used and the collection time-points, independence, validity, reliability, appropriateness to well-being impact questions); (4) analysis (assessment of methodological quality/limitations); (5) results and conclusions; (6) the presence of possible conflicts of interest for authors. In order to capture project details in the grey literature, we used an adapted version of the Public Health England Arts and Health Evaluation Framework[43] and extracted project aims; costs; commissioners, partners and funding sources; intervention details; population and reported outcomes. Where available, evaluation details (aims, objectives, budget, procedures, timeline, data analysis and findings) were also extracted.

Our protocol included for us to contact the authors of articles if the required information could not be extracted and was essential for the interpretation of their results but we did not need to do this.

## Quality assessment

To assess the methodological quality of the included published studies, two review authors independently applied the quality checklist for quantitative studies based on the Early Intervention Foundation checklist and detailed in the What Works Centre for Wellbeing methods guide[44] (online supplementary appendix 4). The checklist was used to indicate if a specific study had been well designed, appropriately carried out and properly analysed. A summary of quality scores is presented in table 2.

We then employed the Grading of Recommendations Assessment, Development and Evaluation working group methodology (GRADE) schema for judging strength and quality of evidence on well-being overall from sport and dance interventions. Four categories of evidence are used in GRADE; high, moderate, low or very low. Applying GRADE, randomised controlled trial (RCT) studies were initially judged as high quality and sound observational studies as low quality. Evidence was downgraded for methodological limitations, inconsistent findings, sparse data, indirect evidence and reporting bias. Evidence was graded upwards if there was a large magnitude of effect or a dose-response gradient. The PHE Arts for Health and Wellbeing Evaluation Framework[43] was used to judge the quality of the grey literature in terms of the appropriateness of the evaluation design, the rigour of the data collection and analysis and precision of reporting.

## Data synthesis

Due to heterogeneity of interventions and well-being outcomes between studies, a meta-analysis was not appropriate. We report the findings narratively. Summaries of the characteristics of the included studies were organised in a tabular form (table 3) and present information on the participants (number and characteristics), intervention and comparison conditions, outcomes and measure used, study design, conclusions and study limitations. Summaries of the results (number of participants, mean scores and SD) for each outcome measure at each measurement point, are presented in table 4 and synthesised in the text according to sport/dance intervention type and well-being outcomes. No studies reported CIs and so these have not been included.

## RESULTS

### Search results

After the removal of duplicates, the electronic searches returned 5597 records for title and abstract screening. Of these, 143 relevant articles remained for full-text assessment as well as 60 additional texts identified through other sources (6 through hand searching the reference lists of included reviews and 54 grey literature submissions were found: 12 received through the call for evidence, 33 via the extended search for grey literature and 9 PhDs found on Ethos). After screening the 203 full texts, 11 studies were included in the systematic review. The search screening process is illustrated in figure 1.

### Study characteristics

The systematic review includes seven RCTs[45–51] (with a total of 884 participants) and one cohort study[52] (93 participants) from the published literature. Three evaluation reports were included from the grey literature. A summary of the characteristics of the included papers is presented in table 3. Table 4 provides a summary of the numerical results for each published study.

The included studies investigated the effects of a range of sport and dance interventions; the most common form of intervention reported were based on meditative practices including yoga[46 50] and Baduanjin Qigong.[48] Other interventions reported included body conditioning, aerobic exercise,[47] dance forms delivered through dance training,[45] hip-hop dance,[47] an empowerment-based exercise intervention programme[49] and specifically identified sports including, body conditioning, and ice skating[47] and Nintendo Wii Active Games.[51] Descriptions of interventions tended to be brief. All studies identified the frequency and type of intervention activity, the duration and content of activity sessions, the delivery site and the number of times per week participants took part. All differed in these characteristics as detailed in table 3. Interventions in six of the RCT studies[45–51] and

**Table 2** Quality checklist scores of included published studies: What Works Centre for Wellbeing checklist

| Authors | Evaluation design | | | | | | | Sample | | | |
|---|---|---|---|---|---|---|---|---|---|---|---|
| | Participants completed the same set of measures before and after intervention | Appropriate random assignment to treatment and control conditions | Group assignment was at the appropriate level (eg, individual, community) | An intent-to-treat design was used | The treatment and comparison conditions are thoroughly described | The extent to which the intervention was delivered with fidelity is clear | Appropriate comparison condition | The sample is representative of the target population and characteristics stated | The sample is sufficiently large to test for the desired impact (min 20 per group) | There is a clear process for determining and reporting drop-out and dose | Overall study attrition no higher than 65% |
| Akandere and Demir[45] | × | × | × | × | × | × | | | × | × | × |
| Amorose et al[52] | × | | × | × | | × | | | × | × | |
| Kanojia et al[46] | × | × | × | | × | × | × | | × | × | |
| Kim and Kim[47] | × | × | × | | | | × | × | × | | × |
| Li et al[48] | × | × | × | × | × | × | × | × | × | × | × |
| Lindgren et al[49] | × | × | × | × | | × | | × | × | × | × |
| Noggle et al[50] | × | × | × | × | × | | × | × | | | × |
| Staiano et al[51] | × | × | × | | × | × | × | × | | × | × |

| Authors | Sample | | | | | | | | | | | Analysis | | |
|---|---|---|---|---|---|---|---|---|---|---|---|---|---|---|
| | Baseline equivalence between treatment and comparison groups | Confounding factors controlled for | Participants blinded to group assignment | Consistent and equivalent measurement | Clear processes for determining and reporting drop-out and dose | Assessed and reported on overall and differential attrition | Appropriate measures were used | Measures used were valid and reliable | Measurement independent of treatment measures | Measurement was blind to group assignment | Included assessment information independent of the participants for example, independent observer | Appropriate methods used to analyse results | Appropriate methods used for the treatment of missing data | Total score: study |
| Akandere and Demir[45] | × | × | × | × | | × | × | × | × | | | × | × | 17 |
| Amorose et al[52] | | × | × | × | × | | × | × | × | | | × | | 9 |
| Kanojia et al[46] | × | | × | × | | | × | × | × | | | × | | 13 |
| Kim and Kim[47] | × | | × | × | | | × | × | × | | | × | | 13 |
| Li et al[48] | × | | × | × | × | × | × | × | × | × | | × | × | 21 |
| Lindgren et al[49] | × | × | × | × | × | × | × | × | × | | | × | × | 19 |
| Noggle et al[50] | × | | × | × | | × | × | × | × | | | × | | 15 |
| Staiano et al[51] | × | | × | × | × | × | × | × | × | | | × | × | 16 |

**Table 3** Characteristics of included studies

Published literature

| Authors Country | Numbers of participants | Participant description | Intervention/comparison | Well-being outcomes and measures used Measurement times | Study design | Limitations (risk of bias) |
|---|---|---|---|---|---|---|
| Akandere and Demir[45] Turkey | n=120 | Gender: 50% female Age: 20–24 years Ethnicity: NR | Dance training intervention 90+20 min warm-up and cool down 3x week for 12 weeks Comparison: no intervention | 1. Depression (Beck Depression Scale) *Before and after 12 week dance intervention* | RCT | ▲ Only one measure used ▲ Small population ▲ Sample already had dance knowledge ▲ Participant details not clearly reported ▲ Baseline levels of depression differ in groups |
| Amorose et al[52] USA | n=93 | Gender: female Age: 13–18 years (M=15.78 years) Ethnicity: mostly Caucasian (90.6%) AI: members of a competitive club volleyball programme in Midwestern United States | Followed a cohort of female adolescent volleyball players through a season of competitive volleyball games. *Approximately 4 months* Comparison: time (before vs after) | 1. Need satisfaction – Sport competence (5-item subscale of the Intrinsic Motivation Inventory) – Need for autonomy (6-item scale: Hollembeak and Amorose 2005) – Need for relatedness (10-item Richer and Vallerand's Feelings of Relatedness Scale) 2. Well-being – Self-esteem (10-item Rosenberg's Self-Esteem Scale) – Burnout (15-item Athlete Burnout Questionnaire) *1–2 weeks before competitive season starts and postseason (1–2 weeks before the last official game/~4 m after start of season)* | Cohort | ▲ Sample bias: one club in Western USA, one sport. All females. Mostly Caucasian – Selection bias: only those that agreed to volunteer. Dropout not reported ▲ Study design: no control group. Only 2 time points looked at Did not assess social contextual factors, eg, coaching behaviour |
| Kanojia et al[46] India | n=50 | Gender: female Age: 18–20 years Ethnicity: NR AI: study conducted in the Department of Physiology, Lady Hardinge Medical College and Smt. Sucheta Kriplani Hospital, New Delhi, India | Yoga *35–40 min 6x week for the duration of three menstrual cycles* Comparison: no intervention | 1. Anger (16-item questionnaire)Trait anxiety (40-item questionnaire) 2. Depression (10-item questionnaire) 3. Subjective well-being (50-item questionnaire) Questionnaires were developed by the Defense Institute of Physiology and Allied Sciences *At the beginning and after completion of three menstrual cycles* | RCT | ▲ Dropout not reported ▲ Recruitment methods not reported ▲ Not possible to double blind ▲ *Consistent findings* |
| Kim and Kim[47] Korea | n=277 | Gender: 48% female Age: 17–22 years (M=20.6 years) Ethnicity: NR AI: Korean high school (n=45) and undergraduate students (n=232) volunteers | One of four exercise sessions: aerobic exercise, body conditioning, hip-hop dancing and ice skating *One-off 40 min session+10 min warm-up and cool down* | 1. Mood (Subjective Exercise Experiences Scale: measuring three dimensions; positive well-being, psychological distress and fatigue) *Before and after the exercise session* | RCT | ▲ Data based on one session only |
| Li et al[48] China | n=222 | Gender: 82.5% female Age: 18–25 years (M=20.78 years) Ethnicity: NR AI: college students recruited from college in China | Baduanjin exercise *1 hour 5x week for 12 weeks* Comparison: usual exercise | 1. Self-esteem (Self-esteem Scale (SES)) 2. Mood/mindfulness (Profile of Mood States (POMS) scale) 3. QoL (WHOQOL-BREF) 4. Stress (Chinese Perceived Stress Scale) 5. Self-symptom intensity (SCL-90 scale) *Baseline (before start), at the end of the intervention (week 13), 12-week follow-up (week 25)* | RCT | ▲ Not blinded ▲ Participants recruited from one medical university ▲ Greater proportion of female participants ▲ Small effect size ▲ *Excellent protocol adherence* ▲ *No significant loss to follow-up* |

Continued

**Table 3** Continued

**Published literature**

| Authors Country | Numbers of participants | Participant description | Intervention/comparison | Well-being outcomes and measures used Measurement times | Study design | Limitations (risk of bias) |
|---|---|---|---|---|---|---|
| Lindgren et al[49] Sweden | n=110 | Gender: female Age: 13–19 years (average=15.3) Ethnicity: NR AI: physically inactive students from secondary schools in low socioeconomic status areas | Empowerment-based exercise intervention programme 45 min moderate exercise+15 min discussion (topics such as healthy lifestyles were addressed) 2x week for 6 months Comparison: waiting list | 1. Self-efficacy (Swedish version of a 10-item General Self-efficacy Scale) 2. Behaviour changes (Social Barriers to Exercise Self-efficacy Questionnaire) Once at the start of the programme and once at end (6 months) | RCT | ▲ Small sample size ▲ High dropout rate |
| Noggle et al[50] USA | n=51 | Gender: 61% female in Yoga group, 47% female in control Age: average age 17 years (grades 11 and 12) Ethnicity: 92.2% white, 3.5% Hispanic, 2.1% African-American, 1.4% multirace and 0.8% Asian AI: students at a public high school in rural western Massachusetts. | A Kripalu-based yoga programme of physical postures, breathing exercises, relaxation and meditation 30 min 2–3x week for 10 weeks (28 yoga sessions total) Comparison: physical education as usual 30–40 min 2–3x week for 10 weeks | 1. Mood (POMS-Short Form) 2. Affect (Positive and Negative Affect Schedule for Children) 3. Stress (Perceived Stress Scale) 4. Positive psychology (Inventory of Positive Psychological Attitudes) 5. Resilience (Resilience Scale) 6. Anger (State Trait Anger Expression Inventory-2TM) 7. Mindfulness (Child Acceptance Mindfulness Measure) One week before and after | RCT | ▲ Small sample size. ▲ Would have been ideal to randomise individually but being in a school setting required allocation at the classroom level ▲ Moderate attendance at the yoga classes |
| Staiano et al[51] USA | n=54 | Gender: 55.6% female Age: 15–19 years Ethnicity: African-American AI: overweight and obese students from an urban public high school | Exergame (EG) intervention—students encouraged to play the Nintendo Wii Active game. Two EG groups: cooperative EG worked with a peer to expend calories and earn points together; competitive EG participants competed against a peer 30–60 min per school day in a lunch-time or after-school programme for 20 weeks Comparison: regular daily activities | 1. Self-efficacy (Exercise Confidence Survey) 2. Self-esteem (Rosenberg Self-Esteem Scale) 3. Peer support (Friendship Quality Questionnaire) Baseline, T2 (10 weeks), T3 (20 weeks) | RCT | ▲ Sample bias: small sample from one school and some attrition |

**Grey literature**

| Authors Country | Participant description | Project/organisation Type of intervention | Evaluation aims and objectives | Study design | Limitations |
|---|---|---|---|---|---|
| Potter and Stubbs[55] UK | n=1498 participated in in-school workshops n=2096 in the final sharing events Age: 11–13 years Participants are from urban and rural areas of deprivation | DanceQuest—dance in school settings (including performing and watching dance such as ballet, contemporary, hip-hop, jazz, street; frequency NR | 1. Examine the processes, outcomes and impacts for both individuals and organisations participating in DanceQuest 2014/2015 – Measure the successes of DanceQuest 2014/2015 against the prescribed aims and objectives established at the outset – Investigate the long-term impacts of DanceQuest 2012/2015 described and presented through representative case studies – Draw out any general lessons for effective practices for future, similar projects delivered by Children & the Arts | Qualitative—interviews, observations and photographs throughout | ▲ Focus on the positive well-being outcomes ▲ Face value reporting used |

Continued

**Table 3** Continued

**Grey literature**

| Authors Country | Participant description | Project/organisation Type of intervention | Evaluation aims and objectives | Study design | Limitations |
|---|---|---|---|---|---|
| BOP Consulting[54] UK | n=23 Age: 8–21 years London (UK) Boroughs of Tottenham and Haringey | *Jackson Lane*—multiarts venue in the community (including contemporary circus, comedy, dance and performance); weekly, time and length NR | Assess the impact of the programme: 1. Who is reached by Jacksons Lane's programmes? 2. What was participants' experience of them? 3. What difference did participating make? | Qualitative—semi-structured interviews with participants and volunteers | ▲ Focus on the positive well-being outcomes ▲ Face value reporting used |
| Mansfield et al[53] UK | Population target: inactive people in the London Borough of Hounslow | *Health and Sport Engagement (HASE) Project*—sport in community settings (including yoga, pilates, swimming, netball, football, adapted and disability sport); weekly 1 hour sessions, 12-month delivery phase | Conduct a longitudinal process evaluation examining the key ingredients of successful HASE community programmes and identify challenges in designing, delivering and evaluating the HASE projects | Qualitative— focus groups, structured observations, in-depth interview methods | ▲ Attempted to search for disconfirming cases and consider the negative well-being impact of sport participation |

AI, additional information; n, number of participants; NR, not reported; M, mean; RCT, randomised controlled trial.

in the cohort study[52] involved delivery by qualified sport or dance instructors in formal group sessions. One RCT used the Nintendo Wii Active Games Programme incorporating a cooperative or competitive peer-to-peer method of participation.[51] A wide range of well-being measures were used and are summarised in online supplementary appendix 5.

Projects reported in the grey literature included the following interventions: martial arts, dance, gym-based exercise, exercise classes, swimming, netball, cycling and football,[53] circus-based skills (eg, juggling, balancing, diabolo)[54] and a range of dance forms.[55] Interventions were led by instructors in group settings. Well-being was evaluated using descriptive statistics and/or thematic analysis from surveys, focus groups, interviews and structured observations.

All of the included studies were carried out in countries categorised in the same group as the UK in the OECD Development Assistance Committee (DAC) categories apart from two (one took place in India,[46] and the other was based in Korea[47]). The sample participants in these two studies were university students, whose educational status indicates their relatively high socioeconomic status, making them broadly comparable with the categorisation of the DAC group in which the UK is located.

### Study quality

The scores for the included studies from the What Works Centre for Wellbeing quality checklist for quantitative data are presented in table 2. The most frequent methodological weaknesses within the studies (with four or fewer studies meeting the criteria) were the absence of an intent-to-treat design, not having a clear process for determining and reporting dropout and dose, not having an appropriate method for the treatment of missing data, not controlling for confounding factors, not being able to blind participants or measurements and not including assessment information independent of the participants. Common (all studies meeting the criteria) strengths included using appropriate measures, independent of treatment measures, giving measures before and after the intervention/control and using appropriate methods to analyse the data. The results of the quality checklist varied across studies, with Amorose et al[52 52] scoring the worst (9 criteria met) and Li et al[48 48] scoring the highest (21 criteria met).

The use of the GRADE schema for judging quality of evidence means that despite the predominance of RCT designs, overall the quality of the published evidence on sport and dance interventions to enhance well-being is low in respect of there being very little evidence in total, methodological limitations noted above, small sample sizes in studies and some sample bias.

Using the PHE Arts for Health and Wellbeing Evaluation Framework, the evidence from the grey literature were judged to have a high degree of credibility. The strongest reports included descriptive and theoretical detail about evaluation methods and acknowledged the

**Table 4** Summary of numerical results of included studies

| Authors | Outcome (measure) | Baseline | | Follow-up 1 | | Follow-up 2 | |
|---|---|---|---|---|---|---|---|
| | | Intervention Numbers Mean (SD) | Control numbers Mean (SD) | Intervention Numbers Mean (SD) | Control Numbers Mean (SD) | Intervention Numbers Mean (SD) | Control Numbers Mean (SD) |
| Akandere and Demir[45] | Depression (Beck Depression Scale) | n=60 15.72 (7.004) | n=60 16.53 (5.922) | n=60 13.90 (5.568)*† | n=60 17.48 (7.740) | N/A | N/A |
| Amorose et al[52] | Need satisfaction; sport competence, need for autonomy, need for relatedness | n=93 Sport competence: 5.71 (0.84) Need for autonomy: 3.79 (0.79) Need for relatedness: 5.47 (1.15) | | n=93 Sport competence: 5.50 (1.07) Need for autonomy: 3.76 (0.59) Need for relatedness: 5.50 (1.21) | | N/A | N/A |
| | Self-esteem (10-item Rosenberg's Self-esteem Scale) | n=93 3.21 (0.45) | | n=93 3.21 (0.47) | | | |
| | Burnout (15-item Athlete Burnout Questionnaire) | n=93 2.05 (0.71) | | n=93 2.15 (0.64) | | | |
| Kanojia et al[46] | Anger (16-item questionnaire) | n=25 Postmenstrual phase: initial cycle 8.84 (4.01) Premenstrual phase: initial cycle 15.0 (5.92)‡ | n=25 Postmenstrual phase: initial cycle 9.12 (4.41) Premenstrual phase: initial cycle 14.32 (5.24)‡ | n=NR Postmenstrual second cycle 7.76 (3.53)§ Premenstrual second cycle 9.52 (4.70)§‡ | n=NR Postmenstrual second cycle 9.04 (4.33) Premenstrual second cycle 14.28 (4.89)‡ | n=NR Postmenstrual third cycle 7.92 (4.29) Premenstrual third cycle 8.52 (4.15)§¶ | n=NR Postmenstrual third cycle 8.96 (4.65) Premenstrual third cycle 13.12 (4.83)‡ |
| | Trait anxiety (40-item questionnaire) | n=25 Postmenstrual phase: initial cycle 40.64 (6.22) Premenstrual phase: initial cycle 46.96 (5.87)‡ | n=25 Postmenstrual phase: initial cycle 41.6 (5.49) Premenstrual phase: initial cycle 46.76 (5.33)‡ | n=NR Postmenstrual second cycle 39.40 (6.69) Premenstrual second cycle 41.48 (5.77)§‡ | n=NR Postmenstrual second cycle 40.24 (6.97) Premenstrual second cycle 45.80 (6.41)‡ | n=NR Postmenstrual third cycle 37.24 (9.14)§¶ Premenstrual third cycle 40.80 (5.75)§¶ | n=NR Postmenstrual third cycle 38.64 (12.76) Premenstrual third cycle 43.88 (7.06) |
| | Depression (10-item questionnaire) | n=25 Postmenstrual phase: initial cycle 6.84 (3.10) Premenstrual phase: initial cycle 10.72 (4.19)‡ | n=25 Postmenstrual phase: initial cycle 6.36 (4.13), Premenstrual phase: initial cycle 9.72 (3.89)‡ | n=NR Postmenstrual second cycle 3.96 (2.59)§ Premenstrual second cycle 5.92 (3.76)§‡ | n=NR Postmenstrual second cycle 6.24 (4.98) Premenstrual second cycle 9.56 (3.22)‡ | n=NR Postmenstrual third cycle 3.12 (2.71)§¶ Premenstrual third cycle 4.76 (2.82)§¶‡ | n=NR Postmenstrual third cycle 6.07 (2.81) Premenstrual third cycle 9.36 (2.96)‡ |
| | Subjective well-being (50-item questionnaire) | n=25 Postmenstrual phase: initial cycle 41.72 (16.05) Premenstrual phase: initial cycle 53.92 (20.35)‡ | n=25 Postmenstrual phase: initial cycle 45.6 (14.05) Premenstrual phase: initial cycle 51.04 (14.89) | n=NR Postmenstrual second cycle 39.64 (16.07)§ Premenstrual second cycle 44.48 (17.87)§‡ | n=NR Postmenstrual second cycle 44.68 (16.5) Premenstrual second cycle 50.40 (18.67) | n=NR Postmenstrual third cycle 37.20 (15.17)§¶ Premenstrual third cycle 40.24 (16.22)§¶ | n=NR Postmenstrual third cycle 43.96 (14.01) Premenstrual third cycle 49.76 (17.02)‡ |

Continued

**Table 4** Continued

| Authors | Outcome (measure) | Baseline Intervention Numbers Mean (SD) | Baseline Control numbers Mean (SD) | Follow-up 1 Intervention Numbers Mean (SD) | Follow-up 1 Control Numbers Mean (SD) | Follow-up 2 Intervention Numbers Mean (SD) | Follow-up 2 Control Numbers Mean (SD) |
|---|---|---|---|---|---|---|---|
| Kim and Kim[47] | Positive well-being (Subjective Exercise Experiences Scale) | Ice skating (n=84): 19 (3.9) Hip-hop dance (n=45): 16.3 (4.2) Body conditioning (n=64): 15.3 (2.9) Aerobic dance (n=84): 16.8 (4.0) | | Ice skating (n=84): 20.4 (3.4) Hip-hop dance (n=45): 19.7 (3.4)* Body conditioning (n=64): 18 (2.8) Aerobic dance (n=84): 19.9 (3.9)* | | N/A | N/A |
| | Psychological distress (Subjective Exercise Experiences Scale) | Ice skating: 8.3 (3.9) Hip-hop dance: 9.8 (4.6) Body conditioning: 10.7 (4.1) Aerobic dance: 9.4 (4.2) | | Ice skating: 8.1 (3.9) Hip-hop dance: 7.3 (4.2)* Body conditioning: 9.6 (3.2) Aerobic dance: 6.7 (2.9)* | | N/A | N/A |
| | Fatigue (Subjective Exercise Experiences Scale) | Ice skating: 10.9 (5.4) Hip-hop dance: 16.2 (4.4) Body conditioning: 15.9 (4.4) Aerobic dance: 14.4 (5.0) | | Ice skating: 13.9 (5.3) Hip-hop dance: 12.9 (4.7)* Body conditioning: 13.9 (4.1) Aerobic dance: 11.2 (4.3)* | | N/A | N/A |
| Li et al[48] | Self-esteem (Self-esteem Scale) | n=101 31.17 (3.69) | n=105 31.41 (3.29) | n=96 (101 included in ITT analysis) 31.56 (3.30) | n=105 (105 included in ITT analysis) 31.31 (3.27) | n=93 (ITT analysis) 30.81 (3.45) | n=101 (ITT analysis) 31.0 (3.71) |
| | Mood/mindfulness (Profile of Mood States (POMS) scale) | n=101 102.3 (16.14) | n=105 103.5 (17.34) | n=96 (101 included in ITT analysis) 106 (15.68) | n=105 (105 included in ITT analysis) 107.4 (17.95) | n=93 (ITT analysis) 103.8 (16.78) | n=101 (ITT analysis) 104.6 (16.89) |
| | QoL (WHOQOL-BREF) | n=101 55.84 (6.65) | n=105 54.94 (6.45) | n=96 (101 included in ITT analysis) 55.09 (6.93) | n=105 (105 included in ITT analysis) 54.26 (7.02) | n=93 (ITT analysis) 56.29 (7.45) | n=101 (ITT analysis) 55.61 (7.45) |
| | Attention (Schulte Grid) | n=101 213.9 (58.84) | n=105 224.6 (47.52) | n=96 (101 included in ITT analysis) 192.4 (47.14) | n=105 (105 included in ITT analysis) 210.4 (54.15)† | n=93 (ITT analysis) 193.9 (54.31) | n=101 (ITT analysis) 202.8 (58.34) |
| | Stress (Chinese Perceived Stress Scale) | n=101 24.22 (5.18) | n=105 23.91 (5.50) | n=96 (101 included in ITT analysis) 23.53 (5.40) | n=105 (105 included in ITT analysis) 22.60 (5.43) | n=93 (ITT analysis) 22.72 (5.72) | n=101 (ITT analysis) 23.22 (5.72) |
| | Self-symptom intensity (SCL-90 scale) | n=101 142.9 (33.58) | n=105 142.1 (32.77) | n=96 (101 included in ITT analysis) 135.6 (31.3) | n=105 (105 included in ITT analysis) 136.2 (32.4) | n=93 (ITT analysis) 130.6 (34.83) | n=101 (ITT analysis) 130.4 (31.94) |
| Lindgren et al[49] | General self-efficacy (General Self-efficacy Scale) | n=55 Median (IQR) 32.0 (11.0–54.0) | n=53 Median (IQR) 32.0 (14.0–47.0) | n=27 Median (IQR) 28.0 (15.0–48.0)*† | n=36 Median (IQR) 35.0 (16.0–48.00) | N/A | N/A |
| | Specific self-efficacy (Social Barriers to Exercise Self-efficacy Questionnaire) | n=56 Median (IQR) Support: 9.0 (3.0–18.0) Social: 22.0 (7.0–35.0) | n=54 Median (IQR) Support: 8.0 (3.0–16.0) Social: 18.5 (7.0–37.0) | n=27 Median (IQR) Support: 8.0 (3.0–17.0) Social: 19.0 (7.0–36.0) | n=36 Median (IQR) Support: 7.0 (3.0–18.0) Social: 19.0 (8.0–31.0) | | |

Continued

**Table 4** Continued

| Authors | Outcome (measure) | Baseline | | Follow-up 1 | | Follow-up 2 | |
|---|---|---|---|---|---|---|---|
| | | Intervention Numbers Mean (SD) | Control numbers Mean (SD) | Intervention Numbers Mean (SD) | Control Numbers Mean (SD) | Intervention Numbers Mean (SD) | Control Numbers Mean (SD) |
| Noggle et al[50] | Mood (POMS-Short Form) | n=36 Mood disturbance (–): 42.8 (19.3) Tension anxiety (–): 6.4 (4.7) Depression-dejection (–): 5.1 (5.0) Anger hostility (–): 6.5 (4.7) Vigour activity (+): 9.8 (4.4) Fatigue inertia (–): 8.3 (5.7) Confusion bewilderment (–): 6.8 (3.5) | n=15 Mood disturbance (–): 44.5 (10.2) Tension anxiety (–): 6.7 (2.8) Depression-dejection (–): 4.9 (3.0) Anger hostility (–): 6.3 (2.7) Vigour activity (+): 10.2 (3.8) Fatigue inertia (–):9.8 (4.5) Confusion bewilderment (–): 6.6 (2.7) | n=35 Mood disturbance (–): 38.4 (16.9)# medium-large effect size=0.689 (Cohen's d) Tension anxiety (–): 5.1 (3.6)# Large effect size=0.870 (Cohen's d) Depression-dejection (–): 4.7 (4.9) Anger hostility (–): 5.7 (5.0) Vigour activity (+): 9.3 (4.0) Fatigue inertia (–): 7.2 (5.2) Confusion bewilderment (–): 6.3 (3.5) | n=15 Mood disturbance (–): 51.2 (20.1) Tension anxiety (–): 9.3 (5.8) Depression-dejection (–): 6.3 (4.2) Anger hostility (–): 7.1 (4.5) Vigour activity (+): 10.9 (3.5) Fatigue inertia (–): 9.3 (4.6) Confusion bewilderment (–): 8.3 (4.1) | N/A | N/A |
| | Stress (Perceived Stress Scale) | n=36 19.2 (7.4) | n=15 19.1 (3.8) | n=35 18.6 (6.2) | n=15 20.3 (5.4) | N/A | N/A |
| | Positive psychology (Inventory of Positive Psychological Attitudes) | n=36 Positive psych attributes (+): 4.5 (1.0) Life purpose/satisfaction (+): 4.7 (1.0) Self-confidence during stress (+): 4.2 (1.0) | n=15 Positive psych attributes (+): 4.5 (0.78) Life purpose/satisfaction (+): 4.8 (0.94) Self-confidence during stress (+): 4.2 (0.67) | n=35 Positive psych attributes (+): 4.5 (1.2) Life purpose/satisfaction (+): 4.8 (1.1) Self-confidence during stress (+): 4.3 (0.98) | n=15 Positive psych attributes (+): 4.2 (0.88) Life purpose/satisfaction (+): 4.6 (0.88) Self-confidence during stress (+): 4.0 (0.90) | N/A | N/A |
| | Resilience (Resilience Scale) | n=36 132.9 (18.4) | n=15 132.1 (12.4) | n=35 131.9 (24.5) | n=15 127.9 (23.4) | N/A | N/A |
| | Affect (Positive and Negative Affect Schedule for Children) | n=36 Positive affect (+): 50.1 (11.5) Negative affect (–): 32.1 (12.5) | n=15 Positive affect (+): 47.7 (9.4) Negative affect (–): 28.8 (7.7) | n=35 Positive affect (+): 48.6 (11.7) Negative affect (–): 29.4 (11.5)# Medium-large effect size=0.659 (Cohen's d) | n=15 Positive affect (+): 49.2 (11.3) Negative affect (–): 38.4 (15.5) | N/A | N/A |
| | Mindfulness (Child Acceptance Mindfulness Measure) | n=36 53.9 (8.6) | n=15 52.3 (6.7) | n=35 53.4 (7.8) | n=15 49.4 (7.2) | N/A | N/A |
| | Anger (State Trait Anger Expression Inventory-2TM) | n=36 Inward (–): 16.4 (4.2) Outward (–): 17.2 (5.7) Control (+): 22.8 (5.5) | n=15 Inward (–): 15.9 (3.3) Outward (–): 16.5 (4.0) Control (+): 22.7 (5.3) | n=35 Inward (–): 16.8 (4.9) Outward (–): 16.9 (5.5) Control (+): 22.4 (6.1) | n=15 Inward (–): 17.9 (4.6) Outward (–): 17.1 (3.7) Control (+): 20.9 (3.7) | N/A | N/A |

Continued

**Table 4** Continued

| Authors | Outcome (measure) | Baseline | | Follow-up 1 | | Follow-up 2 | |
|---|---|---|---|---|---|---|---|
| | | Intervention Numbers Mean (SD) | Control numbers Mean (SD) | Intervention Numbers Mean (SD) | Control Numbers Mean (SD) | Intervention Numbers Mean (SD) | Control Numbers Mean (SD) |
| Staiano et al[51] | Self-efficacy (Exercise Confidence Survey) | Cooperative (n=19): 38.16 (12.12); Competitive (n=19): 36.37 (13.97) | n=16 37.38 (12.07) | Cooperative (n=18): 42.11 (13.58); Competitive (n=17): 37.65 (10.03) | n=14 34.57 (11.75) | Cooperative (n=14): 43.29 (13.40); Competitive (n=11): 38.82 (8.82) | n=10 35.30 (8.76) |
| | Self-esteem (Rosenberg Self-Esteem scale) | Cooperative (n=19): 22.79 (4.45); Competitive (n=19): 23.74 (6.47) | n=16 22.69 (3.96) | Cooperative (n=18): 22.67 (5.91); Competitive (n=18): 23.11 (4.78) | n=15 22.40 (5.38) | Cooperative (n=13): 24.08 (3.88); Competitive (n=9): 22.33 (5.74) | n=11 20.45 (5.82) |
| | Peer support (Friendship Quality Questionnaire) | Cooperative (n=19): 71.89 (12.43); Competitive (n=19): 64.37 (19.58) | n=16 70.13 (18.16) | Cooperative (n=18): 75.22 (13.39); Competitive (n=18): 72.44 (10.78) | n=15 72.33 (17.15) | Cooperative (n=11): 80.18 (8.59); Competitive (n=13): 76.92 (14.04) | n=10 59.70 (20.67) |

*P<0.05 from baseline to follow-up within groups.
†P<0.05 between groups at follow-up.
‡(Kanojia et al (2013)[46] P<0.05 comparison between premenstrual and postmenstrual phase.
§(Kanojia et al (2013)[46] P<0.05 in comparison with initial cycle.
¶(Kanojia et al (2013)[46] P<0.05 in comparison with second cycle.
n, number of participants; N/A, not applicable; NR, not reported; ITT, intention to treat.

limitations of evaluation design. Two studies reported both preproject and postproject data. It was not always clear how themes were identified and developed and it was not always apparent that conclusions emerged from comprehensive data treatment. One report made a clear attempt to search for disconfirming cases and consider the negative well-being impact of sport participation,[53] but evaluation reports tended to focus only on the positive impacts of sport and dance. Furthermore, there was a tendency in evaluations on dance and performance to rely on face value reporting of participants' accounts rather than developing latent forms of thematic analysis informed by identified theory where appropriate.

### The effect of meditative sport activity on well-being

Three published RCT studies assessed the effectiveness of meditative practices including yoga[46 50] and Baduanjin Qigong[48] on well-being in young people. All three studies used several different measures of well-being including mood scales for rating anger, anxiety, positive and negative affect, confusion/bewilderment and stress, anxiety and depression.[46 48 50] One study also included measures of self-esteem, quality of life, mindfulness and resilience.[48] Two studies reported significantly improved well-being outcomes from taking part in yoga compared with a control group.[46 50] One study found significant reductions between groups in total mood disturbance (medium-large effect size=0.689 (Cohen's d), p=0.015), tension and anxiety (large effect size=0.870 (Cohen's d), p=0.002) and negative affect (medium-large effect size=0.659 (Cohen's d), p=0.006).[50] The second study found a significant reduction at 3 months compared with baseline in self-reported depression (effect size=not reported (NR), postmenstrual phase p<0.001, premenstrual phase p<0.001), anxiety (effect size=NR, postmenstrual p<0.05, premenstrual p<0.001), and anger (effect size=NR, premenstrual p<0.001), as well as an improved overall sense of well-being (effect size=NR, postmenstrual p<0.001, premenstrual p<0.001).[46] One study reported no significant difference in self-esteem, mindfulness, quality of life, stress or 'sympton' intensity in young people taking part in Baduanjin Qigong compared with usual exercise practice.[48] No grey literature on yoga and well-being was included in this review.

### The effect of group/team sport on well-being

Two published RCT studies[49 51] and one cohort study[52] examined the well-being outcomes of group sport activities. Two of these studies measured well-being using self-efficacy scales.[49 51] Two studies included a measure of self-esteem.[51 52] One study used a friendship quality assessment as a measure of well-being.[51] One study measured well-being outcomes relating to need satisfaction theory (competence, autonomy and relatedness)[52]; an established approach to personal well-being research in sport psychology. The two studies using self-efficacy measures reported statistically significantly improved scores after taking part in group sport interventions compared with

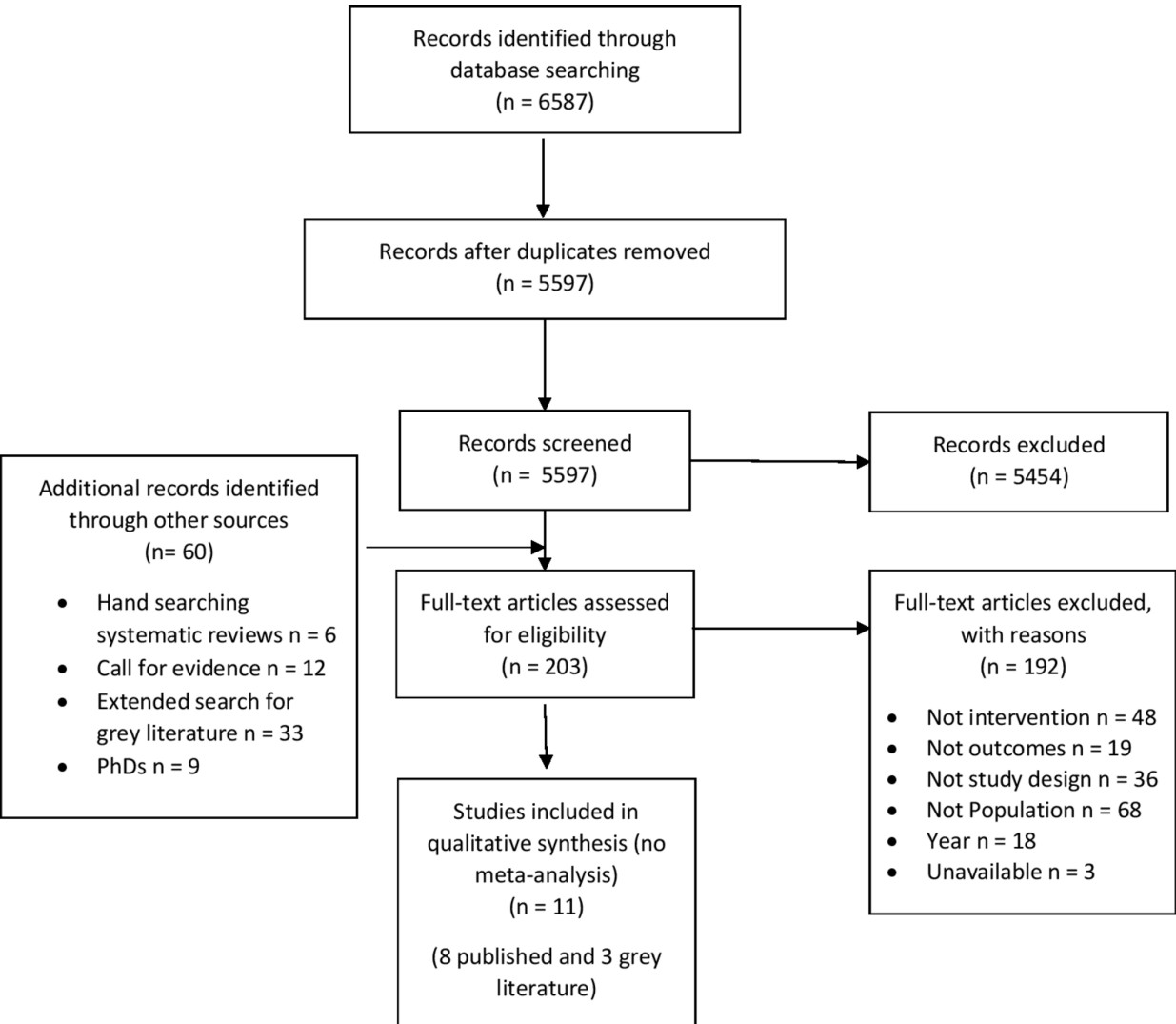

**Figure 1** Preferred Reporting Items for Systematic Reviews and Meta-Analysis flow diagram of the search screening process.

the control (effect size=NR, p=0.037[49]; cooperative condition (M=43.29, SD=13.40) versus control group (M=35.30, SD=8.76), t=2.99, p=0.005).[51]

Both these studies employed interventions that were tailored to the needs of the participants and included elements of peer support. Significant increases in friendship quality were reported in taking part in sport compared with no sport (control condition: M=59.70 SD=20.67; cooperative condition: M=80.18, SD=8.59, t=2.76, p=0.010; competitive condition: M=76.92, SD=14.04, t=3.66, p=0.001).[51] No significant differences were reported for self-esteem scores between sport intervention groups compared with control.[51] Changes in sports players' need to feel competent, autonomous and connected to others over the course of a sporting season were found to be positively related to changes in their overall sense of self-esteem.[52] Qualitative findings from the one grey literature report identified negative and positive aspects of well-being associated with engagement in community sport including enhanced feelings of social connectedness, pleasure and sense of purpose as well as

concerns related to personal capability, competence and unfavourable comparisons to peers who are 'sporty'.[53]

### The effect of group dance on well-being

Two published RCT studies examined the well-being outcomes (mood, fatigue scores and levels of depression) of group dance activities.[45 47] One used a bespoke dance training programme,[45] the other compared dance activities with sport and fitness activities.[47] Taking part in dance exercise to music (aerobics) and hip-hop dancing aerobics were reported to significantly improve self-reported positive well-being and reduce distress and fatigue at the end of the intervention (effect size=NR, p<0.05).[47] Significant improvements on the self-reported Beck Depression Scale (0–9=not depressed; 10–15=low-level depression; 16–23=medium-level depression, 24+=depressive) in participants not diagnosed with depression were reported from a dance training intervention (M=13.90, SD=5.568) compared with control (M=17.48, SD=7.740); t=2.911, p=0.004.[45] The grey literature reported questionnaire and interview results showing positive well-being associations

from dance interventions in terms of increased confidence, sense of purpose and fun and exhilaration.[54 55] Dance was also found to enhance, happiness, relaxation, playfulness, fun, social connectedness, aspiration, ambition and reduce isolation.[54]

## DISCUSSION
### Principal findings and contribution to knowledge

The relationship between organised physical activity and well-being in young people is not well understood. To our knowledge, this is the first systematic review of sport and dance interventions to promote subjective well-being in healthy young people (15–24 years). Overall, the published evidence suggests that meditative physical activity (examples included here were yoga and Baduanjin Qigong) has the potential to improve subjective well-being in terms of reduced anxiety, depression and anger, and enhanced positive mood in young people. This evidence also shows that taking part in dance can lead to positive well-being outcomes in terms of mood enhancement and self-reported reductions in feelings of depression in some youth populations. Unpublished grey literature illustrated that taking part in or watching dance, or other forms of performance-based physical activity and community sport can instil positive well-being feelings such as exhilaration and sense of purpose, and increased confidence, self-esteem and feelings of belonging and purpose. The findings support work that has associated physical activity with positive outcomes connected to depression, anxiety, self-esteem and cognitive function in children and adolescents.[30 56 57] The findings of this review also suggest that group-based sport and dance interventions may be important in ensuring positive well-being outcomes for young people taking part. Research supporting the physical and mental health contributions of physical activity has identified mediators such as organisational practices and the role of seeing other people who are similar to you becoming and being active, which are significant determinants of physical activity engagement.[58] Our evidence also suggests that peer-supported delivery mechanisms in sport and dance programmes may support well-being enhancement for young people. This finding reinforces evidence-based calls for well-designed, clearly focused, expertly led, peer-peer youth interventions which incorporate high-quality peer leader training for positive well-being and mental health outcomes.[59 60] The findings of the unpublished literature suggested that taking part in community sport is also associated with negative well-being connected to concerns about competency and capability. Several studies identify well-being-related and mental health risks in performance-based physical activity for young people including exercise addiction[61 62] and disordered eating linked to feelings of inadequacy and self-criticism.[63] Our findings support work that identifies the need to tailor physical activity

interventions to the needs of those taking part in order to overcome negative perceptions of sport and barriers to involvement in order to maximise the potential for positive well-being outcomes from taking part.[64]

### Implications for policymakers and future research

The findings reported in this review should be treated with caution because the quality of the published evidence on sport and dance interventions to enhance well-being is judged generally to be low. The evidence in this review is sparse, there are methodological limitations in the included studies and we still know very little about the effect of sport and dance interventions, which have the potential to influence the well-being of large numbers of people. No published UK studies were eligible for inclusion in this review. It is not possible to conclude that findings in this review are generalisable across countries or regionally in the UK. The lack of evidence identified in this review does not necessarily mean that well-being benefits are not accrued from taking part in sport and dance. Large-scale community sport and dance interventions have the potential to influence the well-being of people at population level. Recent national sport strategy in the UK[4 5] identifies well-being as an outcome for sport and physical activity and needs to be accompanied by agreement about definitions and measures of well-being, a focus on measuring well-being outcomes and an emphasis on evaluating what works to enhance well-being in sport and dance. National agencies across the sport, culture and health sectors (eg, Department for Digital Culture Media and Sports (DCMS), Arts Counsil for England (ACE), Sport England, Public Health England (PHE) may be influential in promoting this approach; conversely, a lack of national lead may discourage academic and service delivery stakeholders from prioritising this.

Based on the evidence in this study, it is necessary to build evidence on well-being outcomes of sport and dance in healthy young people using agreed measures of well-being. There is a need for more well-designed, rigorous studies which are underpinned by relevant theory. Large-scale randomised controlled designs should be implemented in this target age group. Other rigorous and systematic study designs including evaluation of the complex community context and mechanisms of intervention effectiveness should be considered. The development of a multilevel programme of well-being evaluation training would support key policy and service delivery personnel and researchers in the sport and dance sectors in ensuring rigorous evaluation of interventions.

## CONCLUSION

The evidence overall for the subjective well-being benefits of sport and dance interventions for healthy young people is limited in quality, selective and drawn from varied national and cultural contexts. The current state of the evidence means that it is not possible to identify

a common effect of sport and dance on the subjective well-being of young healthy people or be certain about the influence of such physical activity on peoples' well-being. There are large gaps in our knowledge about the effect of sport and dance on the well-being of young people. Knowledge should be improved through rigorous complex community intervention research incorporating valid comparator groups to determine which sport and dance interventions are most effective in improving well-being in young people. Measurement of quantitative outcomes and evaluation of qualitative processes to determine how such interventions achieve their outcomes is needed.

**Acknowledgements** The authors acknowledge the expert support for the searches provided by library-based information scientists at Brunel University London and the University of Brighton.

**Contributors** The review was conceived and designed, and the protocol developed by LM, TK, CM, LGD, JL, AJ, ND, PD, ST, GJ, AP, AT and CV; article screening was carried out by LM, TK, AJ, LGD, JL and CV; data extraction, quality checks data interpretation were completed by LM, TK, AJ, LGD, JL and CV; and the manuscript drafted by LM and critically reviewed by TK, CM, LGD, JL, AJ, ND, PD, ST, GJ, AP, AT and CV.

**Funding** This study was funded by Economic and Social Research Council (ES/N003721/1).

**Competing interests** None declared.

**Patient consent** Not required.

**Provenance and peer review** Not commissioned; externally peer reviewed.

**Data sharing statement** The appendix is available as online supplementary material and includes; Appendix 1, demonstration Ovid MEDLINE search strategy; Appendix 2, table of excluded studies; Appendix 3, the standardised data extraction form; Appendix 4, the What Works Centre for Wellbeing quality checklist (quantitative studies); Appendix 5, summary of SWB measures used in included studies.

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
