## [Reviewer comments · BMJ Open]

ARTICLE DETAILS

TITLE (PROVISIONAL)	Sport and dance interventions for healthy young people (15-24 years) to promote subjective wellbeing: A systematic review
AUTHORS	Mansfield, Louise; Kay, Tess; Meads, Catherine; Grigsby-Duffy, Lily; Lane, Jack; John, Alistair; Daykin, Norma; Dolan, Paul; Testoni, Stefano; Julier, Guy; Payne, Annette; Tomlinson, Alan; Victor, Christina

VERSION 1 – REVIEW

REVIEWER	patrizia calella parthenope university, naples
REVIEW RETURNED	06-Jan-2018

GENERAL COMMENTS	The authors aimed to describe the subjective wellbeing with sport and dance interventions in healthy young people. This is important and relevant in order to understand the sport and dance impact on wellbeing. However, the review is too redundant on some sections like the methodology and there is not a clear discussion section, which made it difficult to contextualize the review in the scientific literature. Therefore, in order to improve the paper, some minor review is suggested:  1) the box at the beginning of the articles is redundant with the last paragraphs of the discussion 2) revise the order of the tables in the manuscript to be sure that they are immediately after the section they are cited 3) revise all the tables to be sure that the information are in the same order and in the same format 4) explain the acronyms presented in the tables 5) the discussion section needs to be improved with some comparison with other studies in the scientific literature See all details in the attached file - The reviewer also provided a marked copy with additional comments. Please contact the publisher for full details.
--

REVIEWER	Brenda Happell University of Canberra, Australia
REVIEW RETURNED	28-Jan-2018

GENERAL COMMENTS	Thanks for the opportunity to review this paper. It is well written and deals with an important topic. Ways to positively influence the wellbeing of young people is crucial in promoting optimal mental and physical health. I was very pleased to see grey literature included.
--

	The introduction could be strengthened with a stronger rationale for the review. p. 3 is 'worthwhileness' a word? Methods: Suggest a justification is provided for the timespan of the review. Otherwise very comprehensive and rigourous section, Results: Well presented, easy to follow Discussion: This section needs the most work. As written it is more like a summary of the results. These need to be clearly related to the broader literature. What does this all mean? How can this knowledge be utilised? How does it relate to what we already know? I encourage the authors to make these changes and good luck with your future work.
--	--

REVIEWER	Greg Atkinson Health and Social Care Institute, Teesside University, UK
REVIEW RETURNED	07-Feb-2018

GENERAL COMMENTS	I was asked to review this systematic review from the perspective of statistical analysis. However, the authors have stated that the interventions and outcome measures were too variable to undertake a robust meta-analysis. Therefore, there are no real statistical issues to scrutinise in my opinion. I have read the various study descriptions and I do agree that the outcomes in particular are heterogeneous and therefore I do agree that I do not think a meta-analysis is warranted in this particular case.
--

REVIEWER	Ale McConnachie Robertson Centre for Biostatistics University of Glasgow Scotland
REVIEW RETURNED	20-Feb-2018

GENERAL COMMENTS	Mansfield et al report a systematic review of sport and dance intervention to improve wellbeing in health young people. This review considers the statistical aspects of the paper. The paper is well written and tells a coherent story. The authors decide that due to the variability between the studies reported, a meta analysis would not be appropriate. This is fully acceptable. That being the case, there is very little for me to comment on in the paper. As far as I can tell, this is a good example of a narrative systematic review, but this is not my area of expertise.
--

VERSION 1 – AUTHOR RESPONSE

Response to Reviews

Reviewers 3 and 4

Reviewers 3 and 4 provided expert statistical review on the paper and we thank them for their comments. Both agree that our decision not to conduct a meta-analysis due to the variability in the interventions and outcome measures in the studies is fully acceptable.

Reviewer 1

The authors aimed to describe the subjective wellbeing with sport and dance interventions in healthy young people. This is important and relevant in order to understand the sport and dance impact on wellbeing. However, the review is too redundant on some section like the methodology and there is not a clear discussion section, which made it difficult to contextualize the review in the scientific literature.

Therefore, in order to improve the paper, some minor review is suggested:

1) the box at the beginning of the articles is redundant with the last paragraphs of the discussion

Thank you for noting the repetition. We have deleted the text at the end of the article and ensured all information is in the box at the beginning as it is our understanding the box is a requirement for BMJ publications

2) revise the order of the tables in the manuscript to be sure that they are immediately after the section they are cited

Thank you for noting the inconsistency. We have reviewed the position of all tables and moved table 1 to the appropriate place after the section in which it is cited. Table 1 now appears on page 7.

3) revise all the table to be sure that the information are in the same order and in the same format

We agree that consistency in formatting of the table is essential and have reviewed and edited accordingly. It is table 3 (characteristics of included studies) and table 4 (summary of numerical results of included studies) that have been specifically edited to respond to this point. In the revised manuscript we include the corrected tables with no track changes. We have uploaded tables 3 and 4 with track changes showing as separate documents for reviewers to see the edits.

4) explain the acronyms presented in the tables

A clearer key to acronyms is not included in the edited tables

5) the discussion section need to be improved with some comparison with other studies in the scientific literature

We agree that the discussion needed to follow a different format and to compare our findings with other studies. We have developed the discussion and edited ensuring more extensive cross referencing to relevant literature. We have retained the section in implications for policy and practice as this is significant to the systematic review work in this project.

See all details in the attached file

Many thanks for providing very clear points in the attached file for us to follow. We have edited accordingly. We have retained the 1992 definition of sport as it is established and remains the citation used in the sport sector. We have made this clear in the text.

Reviewer 2

Thanks for the opportunity to review this paper. It is well written and deals with an important topic. Ways to positively influence the well-being of young people is crucial in promoting optimal mental and physical health. I was very pleased to see gray literature included.

The introduction could be strengthened with a stronger rationale for the review.
p. 3 is 'worthwhileness' a word?

We agree with the need for a stronger rationale for this important topic. We have edited the end of the introduction to include this text and cross reference to relevant literature.

“Interventions that positively influence the wellbeing of young people have the potential to promote good physical and mental health. [31-33] This review provides evidence that may improve understanding of the effects of sport and dance on a range of SWB measures and contribute to informing policy development, programme delivery and measurement and evaluation of sport and dance interventions to enhance wellbeing”

Worthwhileness is a word; a noun referring to the quality of being worthwhile

Methods:

Suggest a justification is provided for the timespan of the review.
Otherwise very comprehensive and rigorous section,

Many thanks for noting this. We have justified the time span as one which would allow us to reflect current and longer-term work on sport, dance and wellbeing

Results:

Well presented, easy to follow

Many thanks.

Discussion:

This section needs the most work. As written it is more like a summary of the results. These need to be clearly related to the broader literature. What does this all mean? How can this knowledge be utilised? How does it relate to what we already know?

We agree entirely and note that this comment is also made by reviewer 1. The discussion certainly needed to follow a different format and to compare our findings with other studies. We have developed the discussion and edited ensuring more extensive cross referencing to relevant literature. We have retained the section in implications for policy and practice as this is significant to the systematic review work in this project.

I encourage the authors to make these changes and good luck with your future work.

Many thanks for all expert reviews and support for this paper. We have made the changes and uploaded a document that shows the edits. In this revised manuscript we include edited tables (3 and 4) but with no track changes. We have uploaded tables 3 and 4 showing track changes in a different document as part of our response.

VERSION 2 – REVIEW

REVIEWER	Patrizia Calella Parthenope University Naples
REVIEW RETURNED	09-Apr-2018

GENERAL COMMENTS	thanks to the authors for the responses. The new version of the manuscript is more clear and complete, also the tables now are easy to read and well defined. In my opinion there is no need for further revisions
--

REVIEWER	Brenda Happell University of Canberra, Australia
REVIEW RETURNED	22-Apr-2018

GENERAL COMMENTS	Thank you for making the suggested changes. The paper is now greatly improved and makes an important contribution to the literature.
--